# The Dictionary for Double Holography and Graviton Masses in d Dimensions

**Dominik Neuenfeld**

*Perimeter Institute for Theoretical Physics, Waterloo, ON N2L 2Y5, Canada*

`dneuenfeld@pitp.ca`

## Abstract

Doubly-holographic models, also known as Karch-Randall brane worlds, have shown to be very useful for understanding recent developments around computing entropies in semi-classical gravity coupled to conformal matter. Although there cannot be a faithful bulk/brane dictionary, we show that there is a simple dictionary which relates brane fields to subleading coefficients of a near-brane expansion of bulk fields—similar to the case of AdS/CFT. We use this dictionary to find a general formula for the leading order contribution to graviton masses in the $d$ dimensional Karch-Randall braneworld.

# 1    Introduction

Recently, braneworld models [1, 2] have received renewed attention in the context of black hole physics [3–7] as well as cosmology [8–12]. In those models, a $d$-dimensional brane is introduced into $d + 1$-dimensional AdS space. The brane tension is variable and for a certain range of tensions a $d + 1$-dimensional graviton mode localizes (or almost localizes) to the brane. This gives rise to an effectively $d$-dimensional theory of gravity on the brane. Moreover, this theory of gravity is coupled to an effective $d$-dimensional conformal field theory which lives on the union of brane and asymptotic AdS boundary [13, 14].

From the point of view of black hole physics, braneworld models with negative cosmological constant on the brane are a useful tool to better understand the recent calculations of the Page curve for large AdS black holes [15, 16]. Amongst other results, they have lead to the insight that the correct way of computing quantum-gravitational entropies in semi-classical gravity is to use the so-called "island formula" [3].

Braneworlds also have a natural interpretation in the AdS/CFT correspondence [17–19], particularly in the generalization of the correspondence to Interface Conformal Fields Theories (ICFT) as well as Boundary Conformal Field Theories (BCFT) [20, 21]. The holographic dual of CFT defects can be modeled by a brane which sits in (and backreacts on) the AdS bulk geometry. In the case of BCFTs the braneworld is related to an End-Of-The-World brane which regulates the bulk.[1] Here, for concreteness, we will focus on the case of BCFTs. The AdS/BCFT correspondence states that an asymptotically AdS bulk where part of the asymptotic AdS boundary is replaced by an AdS brane, has a dual description in terms of a BCFT, with the boundary located where the bulk brane intersects the asymptotic boundary. In the braneworld literature this construction is known as the Karch-Randall model [2]. AdS/BCFT yields two dual descriptions of the system. On the one hand we have the "boundary perspective" which describes the system by an ambient CFT on a space with boundary, where it couples to a conformal field theory in one dimension less. On the other hand we have a bulk description in terms of an asymptotically AdS spacetime where parts of the asymptotic boundary are regulated by a brane.[2] In the limit where the AdS brane sits very close to the cut-off asymptotic boundary, gravity locally localizes to the brane and the holographic system admits a third description beyond the usual AdS/BCFT duality. This "brane perspective" consists of a CFT which lives on a geometry consisting of two connected pieces. One piece is simply the fixed, non-dynamical geometry of the conformal boundary. The other piece is the geometry of the ETW brane. Here, the CFT is only an effective theory and is coupled to *massive* gravity. The three descriptions are shown in fig. 1. It can be demonstrated

---

[1]Modelling the defect or boundary as a brane is only an effective description. For both cases, smooth solutions to supergravity are known, e.g., [22–27]. See also [28] for a doubly-holographic model in from a top-down perspective.

[2]Note that this brane is at best an effective description. In a fully stringy construction one expects the spacetime to degenerate [29, 30].

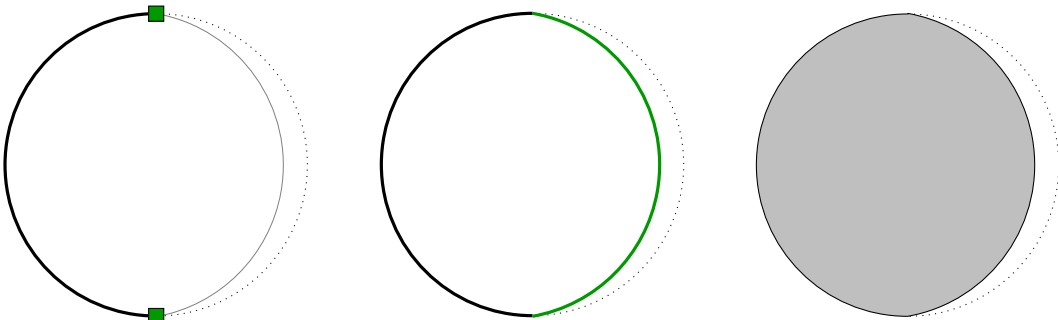

Figure 1: The three descriptions of the braneworld. The metric on the brane is global $AdS_d$. Left: The "boundary perspective" describes the system by an ambient BCFT (solid black) which is coupled to lower-dimensional CFTs on its boundary (dark green), such that the maximal amount of conformal symmetry is preserved. Center: The "brane perspective" is an effective description of the system as a CFT which lives on a non-gravitating (solid black) and gravitational (green) background. Excitations can pass freely between those two regions. Right: The "bulk perspective" descibes the system as an $AdS_{d+1}$ spacetime (shaded) with part of the boundary (dotted line) cut off by the brane.

that the island formula in this third picture is equivalent to the usual RT formula in the $d + 1$-dimensional bulk description [3, 6, 7], where the RT surface is allowed to end on the brane.

Nonetheless, important open questions remain. Unlike in two dimensions [31], in higher dimensions the microscopic origin on the island formula still is unclear. For some recent work on this question see, e.g., [32–34]. Studying doubly holographic models might help elucidate the microscopic origin of the island formula. In fact, at the moment, doubly-holographic scenarios are the only models in higher dimensions in which we have at least some control over the UV definition of a gravitational theory, whilst at the same time having some understanding of how the island formula arises.

It thus seems desirable to have a simple dictionary between the "brane perspective" and the other descriptions of the system in terms of a (B)CFT or asymptotically AdS space with a co-dimension one defect in order to understand the relation between those formulations better.

For the case of the Randall-Sundrum model, in which a flat brane cuts off the entire AdS boundary, aspects of such a dictionary have been discussed in [14, 35, 36]. However, in the case where only part of the boundary is replaced by the brane, like the Karch-Randall model which we will be interested in, new effects like the aforementioned graviton mass appear. This warrants further examination. While the standard holographic dictionary between the $BCFT_d$ and the $AdS_{d+1}$ bulk was discussed early on, see e.g. [37, 38], the literature on holography in the Karch-Randall scenario is rather sparse [39].

Here, we will discuss a version of the dictionary which relates CFT operators to certain coefficients in a near boundary expansion of bulk fields, similar to the well-known result

that the stress-energy tensor is dual to a particular coefficient of the bulk metric in a Fefferman-Graham expansion [40]. As we will see, this also enables us to extract other quantities, such as the graviton mass on the brane [2, 39, 41, 42].

From the onset it is clear that the brane/bulk dictionary we seek cannot be on the same footing as the the usual bulk/boundary dictionary [18, 19]. This is most obvious from the fact that on the brane we only have an effective theory which breaks down at high energies, where the higher-dimensional nature of the bulk becomes visible. This is unlike the BCFT living on the asymptotic boundary, which serves as a definition of the system and is a true conformal field theory. Rather, we should view the brane perspective to be valid only on a small subspace of the full Hilbert space containing low energy fluctuations around a fixed background. Because of this, we will only work in the limit of fluctuations at linear order around a fixed background, i.e., the large $N$ limit.

In standard AdS/CFT there are two equivalent methods to obtain correlation functions on the asymptotic boundary [43]. We can either use the AdS/CFT correspondence to obtain the CFT partition function and extract correlation functions by repeatedly taking functional derivatives with respect to sources located at the boundary [18, 19]. This has been called the "differentiate dictionary". Alternatively, we can extract boundary correlation functions using the "extrapolate dictionary" [44, 45], i.e., by considering bulk correlation functions and taking their limit to the boundary while rescaling by an appropriate function of the holographic coordinate. In braneworld scenarios the first method is not straightforwardly available. The reason is that from the brane point of view the sources become dynamical fields and a variation with respect to the sources vanishes if their equations of motion hold. This is related to the fact that contrary to AdS/CFT bulk fluctuations now describe both, fluctuations of the source as well as associated operators on the brane. Put differently, we need to find a way of telling apart the asymptotic value of the field which is responsible for the sources and the part which is responsible for the operators. This makes the second option seem more helpful. However, as we will see, the extrapolation of a bulk field to the brane has contributions from dynamical sources and only a subleading term in an expansion near the asymptotic boundary which is cut off by the brane corresponds to the expectation value of a brane CFT operator.

The summary of the paper is as follows. In the first part we review the braneworld construction we are interested in as well as basic aspects of holographic renormalization. Based on this we argue that brane observables are given by taking subtracted bulk operators to the brane. The subtraction removes contributions to the field which in a near-boundary expansion take the form of non-normalizable modes. In the case of the stress-tensor we find that to leading order in the limit where the brane approaches the asymptotic boundary (see section 2.3)

$$\langle T_{ij}(x)\rangle = \epsilon_B^{(d-2)/2}\left(\frac{dL}{16\pi G_N^{(d+1)}}g_{(d)ij}(x) + X_{ij}^{(d)}[g^{(0)}]\right) + \mathcal{O}(\epsilon_B^{(d-1)/2}), \quad (1)$$

where $\epsilon_B$ is set by the location of the brane and $g_{(d)ij}(x)$ is a particular term in the

Fefferman-Graham expansion. While eq. (1) looks very similar to the standard expression for the stress-energy tensor, there are important differences. First, we of course have finite corrections as indicated. More importantly, in contrast to the usual AdS/CFT correspondence the bulk metric fluctuations also enter terms in the Fefferman-Graham expansion which are leading relative to $g_{(d)ij}(x)$, so that the stress-energy expectation value is not the leading order contribution anymore as we approach the boundary. However, those leading order terms should be identified with the induced metric on the brane.

The general picture that arises is as follows. The asymptotic behavior of bulk modes as we extend the solutions beyond the brane contain both, finite as well as diverging terms. Roughly speaking, and like in the AdS/CFT correspondence, the former are associated to operator expectation values in the CFT while the latter are associated to sources, which now become dynamical. From the "brane perspective", this means that excitations of the CFT generically mix with excitations of the sources. In the case of the Karch–Randall model gravity, this implies that the new bulk mode which is usually associated with the graviton on the brane is in fact a mix of fluctuations of the brane metric and certain excitations of the stress energy tensor. The well-known effect of this is that the graviton on the brane becomes massive [2].

In the second part of the paper we test this picture. In section 3, we utilize it to study brane graviton masses in $d$ dimensions and at leading order in the near boundary expansion. We find that the mass of the graviton on a $d$-dimensional brane is given to leading order by

$$m_g^2 = \frac{2(d-2)\Gamma(d)}{\Gamma(\frac{d}{2})^2} t^{\frac{d}{2}}, \tag{2}$$

where $t$ is a parameter that depends on the brane tension given in eq. (60). This reproduces the known $4d$ result [41,42]. For higher dimension, the above formula agrees with numerical results.

# 2 Braneworlds and a Bulk-Brane Dictionary

## 2.1 Braneworlds and the AdS/CFT correspondence

We start by reviewing the braneworld construction and its relation to AdS/CFT. Consider an asymptotically AdS space with AdS length $L$ and which preserves an $SO(d-1,2)$ subgroup of $SO(d,2)$ in conformal slicing coordinates

$$ds^2_{\text{AdS},d+1} = \frac{L^2}{f^2(\theta)} \left(d\theta^2 + ds^2_{\text{AdS},d}\right). \tag{3}$$

Here, $ds^2_{\text{AdS},d}$ is a $d$-dimensional AdS metric and $f^{-2}(\theta)$ is a warp factor such that $f(\theta) \to \theta$ as $\theta \to 0$ and $f(\theta) \to (\pi - \theta)$ as $\theta \to \pi$.

We cut off the bulk spacetime at a hypersurface of fixed extrinsic curvature where we introduce a brane. In the above coordinates, fixed extrinsic curvature corresponds to fixed $\theta = \theta_B$ so that the spacetime runs from $\theta = 0$ to $\theta_B$.[3] The brane action is taken to be

$$S^{\text{brane}} = -T_0 \int_{\text{brane}} \sqrt{-\gamma}, \tag{4}$$

where $\gamma_{ij}$ is the induced $\text{AdS}_d$ metric on the brane. This action could be complemented by additional couplings to bulk fields pulled back to the brane, or fields which are localized on the brane. For example one could add an intrinsic Einstein-Hilbert term to the brane [46]. However, we will ignore this possibility here.[4] In order to have a well-defined variational principle, we need to impose boundary conditions at the brane. As is standard, we will assume Neumann boundary conditions which determines the brane tension in terms of $\theta_B$ [1, 2].

The Neumann boundary condition plays the role of an equation of motion for the brane and takes the form

$$K_{ij}[\gamma] - \gamma_{ij}K[\gamma] + 8\pi G_N T_0 \gamma_{ij} = 0. \tag{5}$$

The first two terms arise from varying the bulk Einstein-Hilbert actions with the appropriate GHY term, while the last term comes from the stress-energy tensor associated to the brane,

$$T_{ij}^{\text{brane}} = \frac{-2}{\sqrt{-\gamma}} \frac{\delta S^{\text{brane}}}{\delta \gamma^{ij}} = -T_0 \gamma_{ij}. \tag{6}$$

We can see from eq. (5) that the tension of the brane and the trace of its extrinsic curvature are related by $K = 8\pi G_N \frac{d}{d-1} T_0$. As $\theta_B \to \pi$ the extrinsic curvature approaches $K = d/L$ so that $T_0$ approaches the critical tension

$$T_{\text{crit}} = \frac{d-1}{8\pi G_N L}. \tag{7}$$

In the limit where

$$0 < T_{\text{crit}} - T_0 \ll 1, \tag{8}$$

one of the graviton bulk modes becomes light and its wavefunction is peaked around the brane—it becomes "locally localized" to the brane [2]. This makes it possible to have an effective description of the system where the brane worldvolume hosts a $d$-dimensional

---

[3]Alternatively, one sometimes considers the case where two copies of the above spacetime are glued together across the brane. The one-sided construction we will focus on is the $\mathbb{Z}_2$ orbifold of the two-sided one.

[4]See [6, 7] for how this affects the physics of RT surfaces in the bulk.

theory of gravity. In the case at hand, where the induced metric on the brane is anti-de Sitter, the graviton on the brane is not massless, but has a small mass.[5]

The limit eq. (8) is called the "critical limit", or "near boundary limit" as it corresponds to the case where the brane in the AdS bulk cuts off spacetime very close to the asymptotic boundary. From now on we will always be in the critical limit.

We can learn more about the previous construction by using the AdS/CFT correspondence. From the point of view of AdS/CFT holography, the above braneworld model is constructed by considering a locally AdS bulk and replacing (part of) the near boundary geometry by a co-dimension one brane with an appropriately tuned tension. For the case of an $AdS_d$ brane embedded in $AdS_{d+1}$ this gives rise to three descriptions of the resulting physical system, see fig. 1:

1. The "bulk perspective" describes the system as a $(d+1)$-dimensional asymptotically AdS spacetime. Part of the bulk is cut off by a brane which sits at a bulk slice of constant extrinsic curvature and intersects the asymptotic boundary. At the brane Neumann boundary conditions are imposed.[6]

2. The "brane perspective" is an effective $d$-dimensional description of the low-energy sector of the system in which a CFT lives on spacetime which consists of two parts. One part is the asymptotic boundary. The second part is the brane geometry. In this latter region the CFT is coupled to gravity. CFT modes are free to cross between the two parts of spacetime. This picture breaks down at high energies.

3. From the "boundary perspective" the system is described by a BCFT which lives on the asymptotic boundary. The boundary corresponds to the intersection of the brane with the boundary. This formulation serves as the definition of the system.

Perspectives one and three are the usual dual descriptions of the AdS/CFT correspondence. The second perspective is a new, effective description that arises when the brane tension in the AdS bulk approaches the critical limit.

At the true asymptotic boundary at $\theta = 0$ we can use the usual methods to obtain CFT observables in terms of bulk quantities. To this end one uses the AdS/CFT dictionary which equates the bulk partition function with boundary conditions $\phi_0$ with the generating functional in the CFT, where the $\phi_0$ play the role of sources for the dual operators,

$$\langle e^{i \int O\phi_0} \rangle_{\text{CFT}} = Z_{\text{grav}}[\phi_0]. \tag{9}$$

Taking variations with respect to $\phi_0$ and setting $\phi_0 = 0$, one can then obtain correlation functions of the CFT operator $O$ which is sourced by $\phi_0$. This procedure is not immediately available near the brane, since thanks to the Neumann boundary condition at the brane, the sources are dynamical and are integrated over.

---

[5]This should be contrasted with the massless graviton in flat or de Sitter braneworlds.

[6]This description should be understood as an effective description of an underlying stringy construction.

An alternative, but equivalent prescription in standard AdS/CFT to obtain correlation functions of an operator $O$ of dimension $\Delta$ is to use the extrapolate dictionary. Here, we start by considering bulk-to-bulk correlators of the field $\phi$ dual to $O$. Taking the bulk operators to the asymptotic boundary and rescaling with an appropriate power of $\theta$,

$$\langle O(x) \rangle \sim \lim_{\theta \to 0} \theta^{-\Delta} \phi(x, \theta), \tag{10}$$

we obtain the CFT operator. This dictionary can be applied to bulk correlation functions in order to relate them to CFT correlators.[7] For us, a slightly different version of eq. (10) will be more useful (which can also be derived using eq. (9)). In a near boundary expansion, the dynamical bulk field $\phi(x, \theta)$ takes the form

$$\phi(x, \theta) = \theta^{\Delta} \phi_{(\Delta)} + (\text{higher order in } \theta) \tag{11}$$

and so by comparing to eq. (10) we might identify $\langle O(x) \rangle$ with $\phi_{(\Delta)}$ up to a dimension dependent constant. In the rest of this chapter, we will argue that to leading order this identification still holds true near the brane and can be used to obtain operators in the "brane perspective".

## 2.2 The Boundary Stress-Energy Tensor and Holographic Renormalization in AdS/CFT

Before we establish how bulk fields are related to brane operators, let us start by reviewing a few basic facts about holographic renormalization which will be needed in the following. We will review the basic ingredients using the example of the CFT stress-energy tensor, closely following the discussion in [47].

The definition of the AdS/CFT dictionary is more subtle than it seems from eq. (9). The reason is that already at tree-level the gravity partition function exhibits IR divergences. Holographic renormalization is a prescription that is used to extract a finite bulk partition function. Once a finite bulk partition function for arbitrary boundary conditions $\phi_0$ is obtained, one can use the AdS/CFT dictionary, eq. (9) to obtain (renormalized) CFT correlation functions.

In order to understand the structure of the IR divergences, it is useful write the bulk metric in the Fefferman-Graham expansion near the asymptotic boundary [48],

$$ds^2 = \frac{L^2}{4\rho^2} d\rho^2 + \frac{L^2}{\rho} g_{ij}(\rho, x) dx^i dx^j, \tag{12}$$

---

[7]Equation (10) gives the correlation function of a CFT on hyperbolic space, up to normalization. For a different conformal frame, would need to approach the boundary using a different choice of slicing direction.

where[8]

$$g_{ij}(x, \rho) = g_{(0)ij} + \rho g_{(2)ij} + \cdots + \rho^{d/2} g_{(d)ij} + h_{(d)ij} \rho^{d/2} \log \rho + \mathcal{O}(\rho^{(d+1)/2}). \qquad (13)$$

In the above coordinates, Einstein's equations are given by

$$\rho[2g'' - 2g'g^{-1}g' + Tr(g^{-1}g')g'] + Ric[g] - (d-2)g' - Tr(g^{-1}g')g = 0,$$

$$\nabla_i Tr(g^{-1}g') - \nabla^j g'_{ij} = 0, \qquad (14)$$

$$Tr(g^{-1}g'') - \frac{1}{2}Tr(g^{-1}g'g^{-1}g') = 0.$$

Expanding these equations order by order in $\rho$ enables us to rewrite all $g_{(n)ij}$ for $n < d$ (as well as $h_{(d)ij}$ in even dimensions) in terms of local expressions of $g_{(0)ij}$. However, this expansion does not fully specify the term $g_{(d)ij}$ and further subleading terms.

The fact that we need to specify two tensors to obtain the full perturbative expansion of the metric near the boundary simply reflects the fact that Einstein's equations are second order. In standard AdS/CFT one imposes Dirichlet boundary conditions at the conformal boundary. This fixes $g_{(0)ij}$, but $g_{(d)ij}$ is only constrained to be a symmetric, conserved two-tensor whose trace is restricted in terms of $g_{(0)ij}$. Different choices of $g_{(d)ij}$ correspond to different states of the CFT. The expectation value of the stress-energy tensor in the corresponding states is given by [40, 49]

$$T_{ij} = \frac{dL}{16\pi G_N^{(d+1)}} g_{(d)ij} + X_{ij}^{(d)}[g_{(0)}]. \qquad (15)$$

The last term in this expression only appears in even dimensions and ensures that the trace anomaly of $\langle T_{ij} \rangle$ takes the correct form.

In order to arrive at eq. (15) one uses holographic renormalization. The process starts with regulating the otherwise divergent bulk on-shell action by introducing an IR cutoff $\rho = \epsilon$ near the boundary,

$$S^{\mathrm{reg},\epsilon} = \frac{1}{16\pi G_N} \left( \int_{\rho \geq \epsilon} d\rho\, d^d x\, \sqrt{G}(R - 2\Lambda) + \int_{\rho=\epsilon} d^d x\, 2K \right) \qquad (16)$$

$$= \frac{L}{16\pi G_N} \int d^d x \sqrt{\det(g_{(0)})} \left( \epsilon^{-d/2} a_{(0)} + \cdots + \epsilon^{-1} a_{(d-2)} - \log \epsilon\, a_{(d)} \right) + \mathcal{O}(\epsilon^0), \quad (17)$$

where in the second line we have followed [47] in highlighting the terms which diverge in the $\epsilon \to 0$ limit. Before we can take the limit $\epsilon \to 0$, the divergent terms need to be removed by adding a suitable (local) counterterm action.

---

[8]The term containing $h_{(d)}$ only exists in even dimensions and is related to the conformal anomaly.

The $a_{(n)}$ can be expressed in terms of $g_{(0)ij}$ and its derivatives. Using the divergent terms only, we can define the counterterm action

$$S^{\text{ct},\epsilon} = -\frac{L}{16\pi G_N} \int d^d x \sqrt{\det\left(g_{(0)}\right)} \left(\epsilon^{-d/2} a_{(0)} + \epsilon^{-d/2+1} a_{(2)} + \cdots + \epsilon^{-1} a_{(d-2)} - \log \epsilon \, a_{(d)}\right). \tag{18}$$

The counterterm action can be rephrased in terms of the induced metric

$$\gamma_{ij}(x) = \frac{L^2}{\epsilon} g_{ij}(x, \epsilon) \tag{19}$$

at the regulating surface $\rho = \epsilon$. Rewriting eq. (18) in terms of the induced metric $\gamma$ introduces terms at order $\mathcal{O}(\epsilon^0)$ and higher. However, up to order $\mathcal{O}(\epsilon^0)$, those terms still only depend on $g_{(0)ij}$ and not on $g_{(d)ij}$.[9]

The full holographically renormalizated action is then defined as the sum of the regulated and counterterm actions after removing the cutoff

$$S^{\text{ren}} = \lim_{\epsilon \to 0}(S^{\text{reg},\epsilon} + S^{\text{ct},\epsilon}) \tag{21}$$

and is finite by construction. This is the action that is used in eq. (9). To obtain the stress energy tensor, we must start with the subtracted action

$$S^{\text{ren},\epsilon} = S^{\text{reg},\epsilon} + S^{\text{ct},\epsilon}, \tag{22}$$

i.e., the sum of the regulated and counterterm action at a finite cutoff. First, we define the stress-energy tensor on the regulation surface at $\rho = \epsilon$ as

$$T_{ij}[\gamma] = \frac{-2}{\sqrt{-\gamma}} \frac{\delta S^{\text{ren},\epsilon}}{\delta \gamma^{ij}}. \tag{23}$$

We then take the regulator to zero, while rescaling $T_{ij}[\gamma]$,

$$T_{ij}[g_{(0)}] = \lim_{\epsilon \to 0} \frac{1}{\epsilon^{d/2-1}} T_{ij}[\gamma], \tag{24}$$

to obtain the CFT stress energy tensor. The result is eq. (15). Explicit expressions for $X_{ij}^{(d)}$ can be found in [47].

---

[9]The reason that the precise choice of $g_{(d)ij}$ does not contribute to the counterterm at leading order can be seen as follows. The term $g_{(d)ij}$ can only contribute at order $\mathcal{O}(\epsilon^{d/2})$ if it comes from expressing $g_{(0)ij}$ by $\gamma$ at the level of the cosmological constant term proportional to $\sqrt{\gamma}$ in the counterterm action. However, any change to the term induced by a change in the state yields

$$\delta\sqrt{\gamma} \sim \delta\sqrt{\tilde{g}} = \sqrt{g}\left(1 + \frac{\epsilon^{d/2}}{2} \delta tr(g_{(d)})\right). \tag{20}$$

While the correction appears at the correct order to affect the stress-energy contribution, the trace $tr(g_{(d)})$ is fixed via the equations of motion in terms of $g_{(0)ij}$. Thus if we only change $g_{(d)ij}$, $\delta tr(g_{(d)})$ vanishes.

## 2.3 The Brane Stress-Energy Tensor

Next, we will use the same logic to understand how gravity arises on the brane and which part of bulk gravity fluctuations we should identify with the stress-energy tensor. The difference to the previous discussion is that now we are not at the asymptotic boundary, but instead at a brane near the asymptotic boundary which cuts off spacetime.

Since the brane acts as an IR regulator, we should not add counterterms in order to make the on-shell action finite.[10] However, in the critical limit the analysis of IR divergences in the preceding section still holds true. In particular, if the brane is tuned such that it sits very close to the boundary, the on-shell action can still be expanded in a near boundary expansion and the leading order terms can be expressed as local functions of the induced metric.

The full action is given by $S^{\text{reg},\epsilon_B} + S^{\text{brane}}$, where we assume that the holographic renormalization procedure has already been carried out at the asymptotic boundary at $\theta = 0$. $S^{\text{reg},\epsilon_B}$ is then the renormalized bulk action, regularized by the brane at $\theta = \theta_B$. It is instructive to add and subtract the counterterm action at the brane to the full action and rewrite

$$S^{\text{reg},\epsilon_B} + S^{\text{brane}} = \underbrace{S^{\text{reg},\epsilon_B} + S^{\text{ct},\epsilon_B}}_{S^{\text{matter}}} + \underbrace{S^{\text{brane}} - S^{\text{ct},\epsilon_B}}_{S^{\text{grav}}}. \tag{25}$$

The counterterm action $S^{\text{ct},\epsilon_B}$ captures the dominant terms in a near boundary expansion. The variation of the on-shell action with respect to the induced metric on the brane must vanish. The vanishing is guaranteed by the Neumann boundary condition which plays the role of an equation of motion for the brane. From the left hand side of eq. (25) we find that it reads

$$T_{ij}^{\text{reg}} + T_{ij}^{\text{brane}} = 0. \tag{26}$$

In our convention the stress energy tensors are defined as in eq. (23). In eq. (25) we have indicated that from the brane point of view, $S^{\text{brane}} - S^{\text{ct},\epsilon_B}$ should be thought of as the gravity action on the brane, while the sum of the remaining terms acts as the matter action. This can be seen as follows. From the right hand side of the above equation it is clear that we can also write the equations of motion as

$$8\pi \frac{G_N^{(d+1)}(d-2)}{L}(T_{ij}^{\text{reg}} + T_{ij}^{\text{ct}}) = 8\pi \frac{G_N^{(d+1)}(d-2)}{L}(T_{ij}^{\text{ct}} - T_{ij}^{\text{brane}}), \tag{27}$$

where $T_{ij}^{\text{reg}}$, $T_{ij}^{\text{brane}}$ and $T_{ij}^{\text{ct}}$ denote the variation of $S^{\text{reg},\epsilon_B}$, $S^{\text{brane}}$ and $S^{\text{ct},\epsilon_B}$ with respect to the induced metric on the brane. The prefactor is chosen to give a canonical normalization to the equations below.

---

[10]This becomes even more obvious in models with spacetime on both sides of the brane, where the brane is clearly part of the bulk.

Expressed in terms of the induced metric, the right hand side takes the form [47]

$$
8\pi \frac{G_N^{(d+1)}(d-2)}{L}(T_{ij}^{\text{ct},\epsilon_B} - T_{ij}^{\text{brane}})
$$
$$
= \left( R_{ij}[\gamma] - \frac{1}{2}R[\gamma]\gamma_{ij} - \frac{(d-1)(d-2)}{L^2}\left(1 - \frac{T_0}{T_{\text{crit}}}\right)\gamma_{ij} + \dots \right),
\tag{28}
$$

where the precise structure of terms at higher order in derivatives can be calculated explicitly and depends on the number of dimensions.[11] We see that the counterterm stress energy tensor looks like an Einstein tensor plus higher order corrections from the point of view of the brane. The reason that we have grouped the counterterm stress energy together with the brane stress energy tension is that the latter takes the form of the cosmological constant term. From eqs. (27) and (28) we can read off Newton's constant on the brane and the brane cosmological constant as

$$
G_N^{(d)}L = (d-2)G_N^{(d+1)} \qquad \text{and} \qquad \Lambda^{(d)} = -\frac{(d-1)(d-2)}{L^2}\left(1 - \frac{T_0}{T_{\text{crit}}}\right),
\tag{29}
$$

where the matter stress-energy tensor is identified with the remaining terms,

$$
T_{ij}^{\text{matter}} = T_{ij}^{\text{reg}} + T_{ij}^{\text{ct}} = \frac{-2}{\sqrt{-\gamma}}\frac{\delta S^{\text{ren},\epsilon_B}}{\delta\gamma^{ij}},
\tag{30}
$$

as indicated in eq. (25).

In summary, we find that with this identification, taking the variation of the full action with respect to the induced metric gives the sourceless Einstein equations (coming from $S^{\text{brane}} - S^{\text{ct}}$) and the stress-energy tensor (from $S^{\text{reg}} + S^{\text{ct}} = S^{\text{ren}}$),

$$
R_{ij} - \frac{1}{2}R\gamma_{ij} + \Lambda^{(d)}\gamma_{ij} + \cdots = 8\pi G_N^{(d)}T_{ij}^{\text{matter}}
\tag{31}
$$

The ellipses denotes higher curvature corrections. These are not necessarily small compared to the stress energy and follow from the counterterm action. The matter stress energy tensor, eq. (30), is defined from $S^{\text{ren},\epsilon}$ in analogy with the procedure using the standard AdS/CFT dictionary, eq. (23), up to the fact that the regulator $\epsilon$ is kept finite and we do not rescale.

To obtain an expression for the stress-energy tensor to leading order in $\epsilon$ we can use a simple trick. From eq. (15) we know that in the case of the AdS/CFT correspondence the proportionality constant between $T_{ij}$ and $g_{(d)ij}$ is given by $dL(16\pi G_N^{(d+1)})^{-1}$. However, we are not at the asymptotic boundary, but at the brane. Moreover, we use the induced metric on the brane as our background metric. To correct for this at leading order, we

---

[11]Also note that above solution needs to be revisited for $d = 2$ [6, 7].

can use eq. (24) without taking the limit. We thus conclude that the one point function of stress-energy tensor on the brane at leading order is given by

$$\langle T_{ij}(x)\rangle = \epsilon_B^{d/2-1}\left(\frac{dL}{16\pi G_N^{(d+1)}}g_{(d)ij}(x) + X_{ij}^{(d)}[g^{(0)}]\right), \tag{32}$$

where $g_{(d)ij}$ is the term in the Fefferman-Graham expansion with coefficient $\rho^{d/2}$.

Note that while this expression looks essentially like eq. (15) there is an important difference. While near the asymptotic boundary bulk fluctuations go like $\rho^{d/2-1}$, this is not true near the brane. Here, the Neumann boundary conditions allow the leading order terms to be of order $\rho^{-1}$. This means that in order to extract the stress-energy tensor on the brane from bulk metric fluctuations, we need to ignore all perturbations which are more leading than $g_{(d)ij}(x)$.

Lastly let us mention that of course the choice of counter term is ambiguous, since we can always add a finite term local in the induced metric on the brane to $S^{\mathrm{ct}}$. However, the above split is the natural choice, since $S^{\mathrm{ct}}$ contains all terms to order $\log(\epsilon)$ which depend on the induced metric and $S^{\mathrm{matter}}$ gives rise—at leading order—to the expected stress-energy tensor of a CFT, with the correct conformal anomaly. Here we furthermore assume that a finite term has been added to the counterterm, which removes the coefficient of the logarithmic divergence from the stress energy tensor.

## 2.4   General Brane Fields

The situation for general fields is a straightforward extension of the above argument to the general case for a bulk field $\phi(\rho, x)$. Let us illustrate this using the example of a bulk scalar field. The leading order contributions to the on-shell action near the brane generate a kinetic term for the source $J(x)$ which is identified with the field induced on the brane $J(x) = \phi(\epsilon_B, x)$. In the brane perspective we also have a dual operator $O(x)$ which couples linearly to $J(x)$ and thus sources its equations of motion. The operator expectation value $\langle O(x)\rangle$ is identified with the variation of the subtracted action with respect to the source. Thus, to leading order in the near boundary limit we identify

$$J(x) = \phi(\epsilon_B, x), \qquad\qquad \langle O(x)\rangle = \frac{1}{\sqrt{\gamma(x)}}\frac{\delta S^{\mathrm{ren},\epsilon_B}[J(x)]}{\delta J(x)}, \tag{33}$$

where $\epsilon_B$ is finite and set by the position of the brane. In the critical limit, in order to get around computing $S^{\mathrm{ren},\epsilon_B}[J(x)]$ in eq. (33), we can use the usual Fefferman-Graham near boundary expansion for bulk solutions. By the usual argument it takes the form

$$\phi(\rho, x) \sim \rho^{(d-\Delta)/2}\phi_0(x) + \cdots + \rho^{\Delta/2}\phi_\Delta(x) + \ldots, \tag{34}$$

where $\Delta$ is the scaling dimension of the dual operator in the CFT. Note that due to the change in boundary condition, the mode $\phi_0(x)$ which in vacuum is usually set to

zero by Dirichlet boundary conditions, now is non-zero and fluctuating. However, since the operator $O(x)$ of the effective brane CFT is constructed from the subtracted action, $S^{\mathrm{ren},\epsilon_B}$, the leading contribution will take the form [40]

$$O(x) \sim \epsilon_B^\Delta \left((2\Delta - d)\phi_\Delta(x) + C[\phi_0]\right) + \mathcal{O}(\rho^{\Delta+1/2}), \tag{35}$$

where $C$ is a local functional of $\phi_0(x)$ which can be obtained using holographic renormalization, but can be removed by a choice of counter-terms.

If we can ignore backreaction between scalar fields and gravity we can thus use the following extrapolate-dictionary like procedure to map bulk correlation functions to boundary correlation functions at leading order:

1. Compute the bulk correlator of the bulk field dual to the boundary operator of interest.

2. Move each bulk field in the correlator individually to the brane, dropping all terms which scale as $\rho^\delta$ with $\delta < \Delta$ as one approaches the asymptotic boundary which is regulated by the brane.

It would be interesting to understand how this procedure needs to be modified when backreaction becomes important.

We have discussed in section 2.1 how as the brane is taken towards the asymptotic boundary a new almost-zero mode of the $d + 1$ dimensional graviton appears and plays the role of the massive $d$ dimensional graviton on the brane. An interesting question is whether or not the same happens for general bulk matter. It turns out that the answer depends on the mass of the bulk field.

The mass of the dynamical source $J(x)$ can be read off from the counterterm action for a massive bulk scalar [47],

$$m_J^2 = \frac{(\Delta - d)(2\Delta - d - 2)}{L^2}, \tag{36}$$

where $\Delta$ is the dimension of the CFT operator dual to the bulk field and we consider the case where $\Delta \neq \frac{d}{2} + k$ with positive integer $k$. The dynamical source does not correspond to a particular bulk mode of fixed energy. Rather, bulk modes correspond to combinations of CFT modes and the dynamical source. This is clear from eq. (33) and we will also see this very explicitly in the next section for the example of gravity. However, with Neumann boundary conditions on the brane a new mode appears which has a mass of eq. (36) plus corrections which vanish as $\epsilon_B \to 0$. In the case of a massless bulk field $\Delta = d$ and we find a mode with vanishing mass plus small corrections in the critical limit. In other words, we again find an almost-zero mode, like in the case of gravity. For massive bulk scalars, a new massive field appears on the brane. The bulk mass scale is set by the inverse bulk AdS length $L^{-1}$, which acts as the cutoff scale from the point of view of the brane. Thus, unless the bulk mass is tuned to small values, the dynamical source on the brane will have a mass

above the cutoff scale. The case of $(d+1)$-dimensional bulk fields with negative mass-squared (but above the Breitenlohner–Freedman bound) seems to lead to an inconsistent situation in which the mass of the brane source violates the BF bound in $d$ dimensions. The basic problem is that the dimensionless product of the source mass-squared and the brane curvature scales as

$$\ell^2_{\text{brane}}m_J^2 \sim -\frac{1}{(\pi - \theta_B)^2} \sim -\frac{1}{\epsilon_B}. \tag{37}$$

This needs to be bounded below by $-\frac{(d-1)^2}{4}$. But it is easy to see from eq. (37) that if eq. (36) is negative, we can always choose some $\theta_B$ such that the BF bound is violated. Equation (36) becomes negative for masses

$$-\frac{d^2}{4} + 1 < m_{\text{bulk}}^2 < 0. \tag{38}$$

Interestingly, this seems to leave bulk masses in the window $-\frac{d^2}{4} \leq m_{\text{bulk}}^2 < -\frac{d^2}{4} + 1$ a consistent choice. This window corresponds to the range of masses where the bulk scalar can be quantized with either Dirichlet or Neumann condition at the asymptotic boundary. Outside this window, it might be that mixing of the source $J$ with operators $O$ increases the mass above the allowed value. We will leave this question for future work.

Bulk gauge fields behave similarly to the case of gravitons. A new almost-zero mode appears and gives rise to a massive vector field on the boundary.[12] Again, the small masses for the almost-zero modes appears because the (naively massless) dynamical source mixes with CFT operators and thereby acquire a mass.

## 3  Graviton Masses and Braneworlds

An important example of a doubly-holographic model is the original Karch-Randall construction. Here, one considers AdS vacuum cut off by a brane which sits at constant extrinsic curvature. That is, we consider the metric eq. (3) with $f(\theta) = \sin(\theta)$ such that locally the geometry looks like the AdS vacuum. The asymptotic boundary is at $\theta = 0$ and the brane is located at some $\theta_B < \pi$.

It is well established that in this case the localized graviton mode acquires a mass [2] and several different ways for computing the mass are known [39, 41, 42]. Nonetheless—at least to our knowledge—this calculation has only been performed in four brane dimensions. Below, we will use the dictionary established in the previous section to give yet another way of computing the graviton mass and derive a general formula for the leading order contribution in the near boundary limit in $d$ brane dimensions. As expected, the result

---

[12]This is very different to the case of a Randall-Sundrum brane, where spin-one fields do not localize to the brane [50]. In the Karch-Randall model this is possible, since—unlike in the Randall-Sundrum case—the kinetic term for the gauge fields induced on the brane is finite.

reproduces the known value in four dimensions as well as numerically computed graviton masses in higher $d$.

## 3.1 General Considerations

From the bulk perspective, graviton masses appear on the brane since the boundary condition at the asymptotic and the brane mixes "normalizable" and "non-normalizable" bulk graviton modes into a new set of normalizable modes. Let us be more precise. The linearized Einstein equations around an asymptotically AdS space allow for two independent types of asymptotic behavior. As we extend the solution beyond $\theta_B$ towards $\theta = \pi$, it may contain terms that diverge $\mathcal{O}((\pi - \theta_B)^{-2}) \sim \mathcal{O}(\rho^{-1})$, while the other possible asymptotic solution is of order $\mathcal{O}((\pi - \theta)^{d-2}) \sim \mathcal{O}(\rho^{d/2-1})$. By analogy to the usual AdS/CFT correspondence, and a slight abuse of notation, we will denote the former behavior as the "non-normalizable" and the latter as the "normalizable" solution. We put their names in quotation marks since in fact the presence of the brane ensures that no divergence appears, i.e., both solutions are normalizable.

So far, our definition does not specify the "non-normalizable" solution completely, since we can always add a "normalizable" solution to a "non-normalizable" to obtain a different "non-normalizable" solution. The correct definition of the linearly independent "non-normalizable" is sensitive to the number of dimensions. In odd dimensions, we define the "non-normalizable" solution to be the solution for which the near boundary expansion has no term of order $\mathcal{O}(\rho^{d/2-1})$. Since the CFT on the brane is defined on AdS space, in even dimensions it has an anomalous stress-energy tensor whose contributions come from the "non-normalizable" mode and contribute at the same order as those of the "normalizable" one. In even dimensions, we require that the "non-normalizable" fluctuation at leading order reproduces the *vacuum* stress-energy tensor of the CFT on the brane background perturbed by the mode. That is, if the vacuum stress energy on the brane contributes $T_{ij}^{\text{vac}} = c\,\gamma_{ij}$ to the cosmological constant, the stress energy in the presence of a graviton excitation gets corrected by the corresponding linearized contribution, $\delta T_{ij}^{\text{vac}} = c\,\delta\gamma_{ij}$. This requirement ensures that a purely "non-normalizable" fluctuation gives a massless graviton.

As we will see below, the "non-normalizable" solution of lowest energy gives the main contribution to the graviton on the brane. The admixture of a "normalizable" piece gives a stress-energy contribution proportional to the induced metric which can be interpreted as a graviton mass. This is different than the situation for de Sitter or Minkowski branes where the brane metric fluctuations simply correspond to a bulk "non-normalizable" mode and consequently, the graviton is massless [1,2].

In order to make contact with section 2 we will use a Fefferman-Graham expansion, where $\rho = 0$ corresponds to the (inaccessible, and thus imagined) asymptotic boundary at $\theta = \pi$ which is excised by the brane. The location of the brane, $\theta_B$, is mapped to $\rho = \epsilon_B$. The true asymptotic boundary at $\theta = 0$ is located at $\rho \to \infty$. For general $\theta$ the relation

between conformal slicing and those Fefferman-Graham coordinates is given by

$$\sin(\theta) = \frac{2\sqrt{\rho}}{\rho+1}, \qquad\qquad \cos(\theta) = \frac{\rho-1}{\rho+1}. \qquad (39)$$

In those coordinates, our ansatz for a bulk graviton is

$$G_{ij}(\rho, x) = \bar{G}_{ij}(\rho, x) + \delta G_{ij}(\rho, x), \qquad (40)$$

where $G_{ij}$ is related to the metric in the Fefferman-Graham expansion, eq. (12) as $G_{ij} = \frac{L^2}{\rho} g_{ij}$, $\bar{G}_{ij}$ is the background metric and

$$\delta G_{ij}(\rho, x) = \sum_{k=0}^{\infty} \left( \alpha_k \psi_k^d(\rho) + \beta_k \psi_k^n(\rho) \right) h_{ij}^k(x) \qquad (41)$$

is a linear perturbation for which we have chosen an ansatz of separation. Since we are interested in the mode that turns into the brane graviton, we will also assume that $\delta g_{ij}$ is traceless and transverse in $d$ dimensions. The scaling ambiguity which comes from a simultaneous rescaling of $h_{ij}^k$ and $\alpha_k, \beta_k$ can be fixed by assuming that $\delta G(\epsilon_B, x) = \sum_{k=1}^{\infty} h_{ij}^k(x)$. The induced metric on the brane is given by $\gamma_{ij}(x) = G_{ij}(\epsilon, x)$.

The functions $\psi_k^d(\rho)$ and $\psi_k^n(\rho)$ correspond to the "non-normalizable" (divergent) and "normalizable" solutions in $\rho$ direction, respectively, as discussed above. In a near boundary expansion around the brane, the functions take the general form

$$\psi_k^d(\rho) = \frac{1}{\rho} + \mathcal{O}(1), \qquad\qquad \psi_k^n(\rho) = \rho^{d/2-1} + \mathcal{O}(\rho^{\frac{d}{2}}). \qquad (42)$$

This also defines our normalization convention.

Naively, $k$ is a continuous index. In our situation, however, we impose Dirichlet boundary conditions at the asymptotic boundary at $\rho = \infty$. This requirement together with the boundary condition at the brane at $\epsilon_B$ fixes the $\alpha_k$ of eq. (41) in terms of $\beta_k$ and quantizes the allowed values of $k$. For the moment we will consider the solution with general $\alpha_k, \beta_k$ leaving them unconstrained, before fixing them later on.

## 3.2   A General Formula

The $h_{ij}^k$ are eigenfunctions of the linearized gravity equations on the brane with eigenvalues $E_k$. As discussed before they take the form of the linearized Einstein equations plus higher order corrections. The mode that localizes to the brane and plays the role of the graviton is the mode with the smallest $E_k \equiv E_0$. Thus, in the following we will focus on that mode and drop the sub-/superscript $k = 0$.

The linearized graviton on the brane obeys the equation

$$\left( \Box + \frac{2}{\ell_B^2} + \dots \right) \delta\gamma_{ij} = -16\pi G_N^{(d)} \delta T_{ij}(x), \qquad (43)$$

with the brane AdS length $\ell_B = L \sin \theta_B$. Here, $\gamma_{ij}$ is the induced metric on the brane and the stress-energy tensor can be computed using eq. (32). We also have allowed for a fluctuation of the stress-energy tensor which sources the equation for the massless graviton. According to our explanation above, the graviton mass arises, because the zero mode $h_{ij}$ contributes not only to fluctuations of the induced metric $\gamma_{ij}$ but also to fluctuations of the stress-energy tensor $T_{ij}$. To obtain the contribution of eq. (41) to the stress-energy tensor fluctuation we can now use our dictionary.

Recall that in odd dimensions the full contribution to $\delta T_{ij}$ only comes from the normalizable fluctuation. In even dimensions, we have contributions from both the "non-normalizable" and "normalizable" modes as well as a contribution due to $\delta X_{ij}$. As explained above, terms coming from the "non-normalizable" fluctuation and $\delta X_{ij}$ only depend on $g_{(0)ij}$ and combine to give a contribution proportional to the vacuum stress energy contribution to the cosmological constant and do not contribute to the graviton mass. In a slightly different form, this observation has also been made in [39], where it was argued that in four dimensions the Riegert term of the CFT effective action does not contribute to the graviton mass.

An easy way to see this is that in models where the brane is flat or de Sitter, metric fluctuations are only proportional to the "non-normalizable" piece and give rise to a massless graviton. Thus, the mass is generated by additional "normalizable" contributions to the stress-energy tensor. This contribution can be obtained by taking the "normalizable" component of eq. (41), take it to the brane, and multiply it by an appropriate proportionality constant, eq. (32). We can read off the leading order "normalizable" component from the expansion eq. (41). At leading order it is given by

$$\delta G^{\text{norm}}_{(d)ij}(x) = \beta h_{ij}(x) = \frac{\beta}{\alpha} \delta \gamma_{ij}(x). \tag{44}$$

Thus we find that if $\beta \neq 0$, the zero-mode indeed contributes a new term to the brane stress-energy tensor,

$$\delta T^{\text{norm}}_{ij}(x) = \frac{\beta}{\alpha} \frac{d \epsilon^{d/2}}{16 \pi G_N^{(d+1)} L} \delta \gamma_{ij}(x). \tag{45}$$

Plugging this result into eq. (43) and using the relation between the $(d+1)$ and $d$-dimensional Newton's constant, eq. (29), we find

$$(\Box + \frac{2}{\ell_B^2} + \dots) \delta \gamma_{ij} = -d(d-2) \frac{\beta}{\alpha} \frac{\epsilon^{d/2}}{L^2} \delta \gamma_{ij} + \mathcal{O}(\epsilon^{(d+1)/2}), \tag{46}$$

where we have combined the contribution due to the "non-normalizable" mode in even dimensions with the ellipsis on the left. From this we can immediately read off the graviton mass in term of the coefficients $\alpha$ and $\beta$ of the near boundary expansion around $\theta = \pi$,

$$m_g^2 = -d(d-2) \frac{\beta}{\alpha} \frac{\epsilon^{d/2}}{L^2} + \mathcal{O}(\epsilon^{(d+1)/2}). \tag{47}$$

Computing numerical values for the graviton mass now reduce to determining $\alpha$ and $\beta$. Once the solution for a bulk graviton is known near the boundary, those coefficients can be read off by splitting the solution into a "normalizable" and a "non-normalizable" contribution, according to our definitions above. The coefficient $\alpha$ is given by the leading order coefficient of the "non-normalizable" solution, while $\beta$ is given by the $\rho^{d/2}$ coefficient of the "normalizable" solution. This is what we will turn to now.

## 3.3   Graviton Masses in the d-dimensional Karch-Randall Model

It turns out to be convenient to solve the equations for the bulk fluctuation in conformal slicing first, then impose the Dirichlet boundary condition at $\theta = 0$, and only then solve for the mass. We again choose an ansatz of separation for a transverse-traceless bulk metric fluctuation in axial gauge

$$\delta G_{ij}(\theta, x) = \frac{L^2}{\rho} \delta g_{ij}(\theta, x) = c\, \chi(\theta) h_{ij}(x). \tag{48}$$

where $h_{ij}(x)$ is proportional to the graviton wavefunction on the brane and is an eigenfunction of the kinetic operator for spin-2 fluctuations on constant $\theta$ slices,

$$\left( \frac{1}{2}\tilde{\Box} + \frac{1}{\ell_B^2} + \dots \right) h_{ij}(x) = \frac{E_0^2}{\ell_B^2} h_{ij}(x), \tag{49}$$

with eigenvalue $E_0$, which is the smallest eigenvalue allowed by the bulk boundary conditions and $\tilde{\Box}$, the tensor Laplacian on AdS with radius $\ell_B$. $E_0$ is proportional to the graviton mass in units of the brane AdS length. As we can see from eq. (47), this means that $E_0^2$ will be of order $\mathcal{O}(\epsilon^{d/2-1})$ in the near boundary limit $\epsilon \ll 1$. Along the slicing direction the function $\chi(\theta)$ obeys

$$\chi''(\theta) - (d-5)\cot(\theta)\chi'(\theta) + \big(2(d-3) - 2(d-2)\csc^2(\theta) + E_0^2\big)\chi(\theta) = 0 \tag{50}$$

This equation can be solved exactly using associated Legendre functions. The general solution is

$$\chi(\theta) = c_1\,(\sin\theta)^{\frac{d-4}{2}}\,P_\nu^{\frac{d}{2}}(\cos\theta) + c_2\,(\sin\theta)^{\frac{d-4}{2}}\,Q_\nu^{\frac{d}{2}}(\cos\theta) \tag{51}$$

with

$$\nu = \frac{\sqrt{(d-1)^2 + E_0^2} - 1}{2}. \tag{52}$$

As previewed above, the anomalous contribution to the stress-energy tensor does not contribute to the graviton mass and thus extracting the graviton mass works completely analogously for both odd and even dimension. We will therefore only show the derivation

for the slightly more complicated case of even dimensions explicitly and only comment on the differences to the simpler, odd dimensional case.

We will start by imposing the Dirichlet boundary condition at $\theta = 0$. This sets $c_2 = 0$ ($c_1 = 0$ for odd dimensions). The value of the unrestricted constant is immaterial. We can therefore simply set $c_1 = 1$. Given the solution $\chi(\theta)$, we can now bring it into the form of eq. (41) by rewriting it as [51]

$$
\begin{aligned}
\chi(\theta) &= (\sin\theta)^{\frac{d-4}{2}} \, P_\nu^{\frac{d}{2}}(\cos\theta) \\
&= (\sin\theta)^{\frac{d-4}{2}} \left[ (-1)^{\frac{d}{2}} \cos(\nu\pi) P_\nu^{\frac{d}{2}}(-\cos\theta) - \frac{2}{\pi}(-1)^{\frac{d}{2}} \sin(\nu\pi) Q_\nu^{\frac{d}{2}}(-\cos\theta) \right].
\end{aligned}
\tag{53}
$$

The benefit of working with associated Legendre functions is that this split corresponds to the split of the solution in eq. (41), where $Q_\nu^{d/2}(-\cos\theta)$ is proportional to the "non-normalizable" and $P_\nu^{d/2}(-\cos\theta)$ is proportional to the "normalizable" contribution. The latter can easily be seen by using eq. (39) to expand $P_\nu^{d/2}(-\cos\theta)$ around $\rho = 0$.

In even dimensions, showing that $Q_\nu^{d/2}(-\cos\theta)$ agrees with our definition of "non-normalizable" mode requires some more work. We can expand eq. (53) around $E_0 = 0$. In even dimensions, $\nu = \frac{d-2}{2} + \mathcal{O}(E_0^2)$ is an integer plus a small correction linear in $E_0^2$. This means that at zeroth order in $E_0^2$, $\sin(\nu\pi)$ in the last term in eq. (53) vanishes and consequently, at first order in $E_0^2$, we can replace $Q_\nu^{\frac{d}{2}}$ by $Q_{\frac{d-2}{2}}^{\frac{d}{2}}$. One can show that

$$
(\sin\theta)^{\frac{d-4}{2}} Q_{\frac{d-2}{2}}^{\frac{d}{2}}(-\cos\theta) \sim \frac{1}{\sin^2\theta},
\tag{54}
$$

and thus to the order of $E_0$ we are interested in, the "non-normalizable" contribution coming from $Q_\nu^{d/2}(-\cos\theta)$ is only a small change in the background

$$
\frac{L^2}{\sin^2\theta}(d\theta^2 + (\bar{g}_{ij}(x) + \delta g_{ij}^d(x))dx^i dx^j).
\tag{55}
$$

A computation of the leading contribution to the stress-energy tensor gives the stress energy tensor induced by the conformal anomaly, now on the deformed background. We have thus established that the coefficients $\alpha$ and $\beta$ in eq. (47) correspond to the leading coefficient of the term proportional to $Q_\nu^{d/2}$ and the coefficient of $\rho^{d/2}$ in the term proportional to $P_\nu^{d/2}$, respectively, in an expansion in Fefferman-Graham coordinates, eq. (39).

Note that in odd dimensions the above argument is not necessary. In this case the role of $P_\nu^{d/2}$ and $Q_\nu^{d/2}$ is exchanged, such that $Q_\nu^{d/2}$ gives the normalizable solution. A simple expansion of $P_\nu^{d/2}$ is sufficient to show that it does not produce a leading order contribution to the stress-energy tensor. This is of course consistent with the fact that in odd dimensions the trace anomaly is absent.

We can now extract the leading terms by using an asymptotic expansion at $\rho = 0$. For

| $d$ | 4 | 5 | 6 | 7 | 8 | 9 | 10 |
|---|---|---|---|---|---|---|---|
| $\frac{m_g^2 L^2}{(\pi-\theta_B)^d}$ | $\frac{3}{2} = 1.5$ | $\frac{8}{\pi} \simeq 2.55$ | $\frac{15}{4} = 3.75$ | $\frac{16}{\pi} \simeq 5.09$ | $\frac{105}{16} \simeq 6.56$ | $\frac{128}{5\pi} \simeq 8.15$ | $\frac{315}{32} \simeq 9.84$ |

Table 1: Graviton mass on a $d$ dimensional brane as a function of brane location $\theta_B$.

integer order the asymptotic behavior of the Legendre functions is given by [51]

$$P_\nu^\mu(-\cos\theta) \sim -\frac{\Gamma(\mu-\nu)\Gamma(\mu+\nu+1)\sin(\pi\nu)}{\pi\Gamma(\mu+1)}\rho^{\mu/2} \tag{56}$$

$$Q_\nu^\mu(-\cos\theta) \sim \frac{(-1)^\mu\Gamma(\mu)}{2}\rho^{-\mu/2}. \tag{57}$$

Using these expression, the ratio $\frac{\beta}{\alpha}$ can be computed as

$$\frac{\beta}{\alpha} = \frac{\cos(\nu\pi)\Gamma(\frac{d}{2}-\nu)\Gamma(\frac{d}{2}+\nu+1)}{(-1)^{\frac{d}{2}}\Gamma(\frac{d}{2})\Gamma(\frac{d}{2}+1)} = -\frac{2}{d}\frac{\Gamma(d)}{\Gamma(\frac{d}{2})^2} + \mathcal{O}(E_0^2) \tag{58}$$

The calculation for an odd number of brane dimensions follows along very similar lines and yields the same result.

Plugging this into eq. (47) we obtain that

$$m_g^2 = \frac{2(d-2)\Gamma(d)}{\Gamma(\frac{d}{2})^2}\frac{\epsilon_B^{\frac{d}{2}}}{L^2} + \dots. \tag{59}$$

Of course, the result can also be written using the beta function $B(\frac{d}{2}, \frac{d}{2}) = \frac{\Gamma(\frac{d}{2})\Gamma(\frac{d}{2})}{\Gamma(d)}$. A coordinate invariant reformulation of the above result can be obtained by expressing $\epsilon_B$ in terms of the brane tension. To this end, let us introduce a dimensionless order parameter $t$ defined as

$$t = \frac{T_{\text{crit}} - T_0}{T_{\text{crit}} + T_0} \sim \frac{T_{\text{crit}} - T_0}{2T_{\text{crit}}}. \tag{60}$$

Using the relation between the extrinsic curvate and the tension discussed around eq. (7) one can show that in fact this order parameter equals $\epsilon_B$ and thus

$$m_g^2 = \frac{2(d-2)\Gamma(d)}{\Gamma(\frac{d}{2})^2}\frac{t^{\frac{d}{2}}}{L^2} + \dots. \tag{61}$$

We consider this the main result of this section. The analytic results for the graviton mass in the near boundary limit are summarized in table 1.

## 3.4 Numerical Results

In [41] the mass of the graviton for $d = 4$ was computed.[13] After converting to our conventions[14], the author found that in conformal slicing coordinates the graviton mass in the near boundary limit is

$$m_g^2 = \frac{3}{2} \frac{(\pi - \theta_B)^4}{L^2}.$$ (62)

As a first check we should compare this to our general formula, eq. (61). In fact, eq. (59) has a more convenient form. From eq. (39) we obtain that in the near boundary region

$$\epsilon_B = \frac{(\pi - \theta_B)^2}{4}.$$ (63)

Substituting $d = 4$ and the above expression for $\epsilon$ into eq. (59) we find perfect agreement with eq. (62).

To our knowledge, the $d = 4$ case is the only result for which an explicit expression has been obtained in the literature. We will thus resort to numerics to check the validity of eq. (61) for the cases of higher dimensions. We do this by numerically integrating the equation for the wavefunction in $\theta$ direction starting at the brane, where we impose the boundary condition

$$\partial_\theta \psi_k(\theta) + 2 \cot \theta \, \psi_k(\theta)|_{\theta = \theta_B} = 0.$$ (64)

This is done for different values of $k$. We then find the values of $k$ for which the wavefunction vanishes at the asymptotic boundary, $\theta = 0$. The smallest positive value of $k$ then yields the graviton mass. Figure 2 shows the graviton mass in units of the AdS length on the brane for $d = 5, 6, 7$. Figure 3 displays the graviton mass divided by $(\pi - \theta_B)^{d-2}$. As one can see from the plots, the analytic near brane result derived in this paper is good for $\theta_B < \pi - 0.1$.

## 4 Discussion

In this paper we have discussed how an extrapolate-dictionary-like procedure can be used to compute braneworld correlation functions from bulk data. This dictionary relates bulk operator expectation values to expectation values in the effective Hilbert space in the lower-dimensional brane theory and it is tempting to speculate that this extends to the operator level within a suitable regime of validity.

This dictionary allows for a simple argument against firewalls [52] for non-generic states. In [7] it was discussed how to use doubly-holography to model a $d$-dimensional eternal black

---

[13]See also [42] for a toy model which gives the same result.

[14]We have that in the near boundary limit Miemiec's $x_0$ becomes $L$, and $-\Lambda$ becomes $(\pi - \theta_B)^2/L^2$.

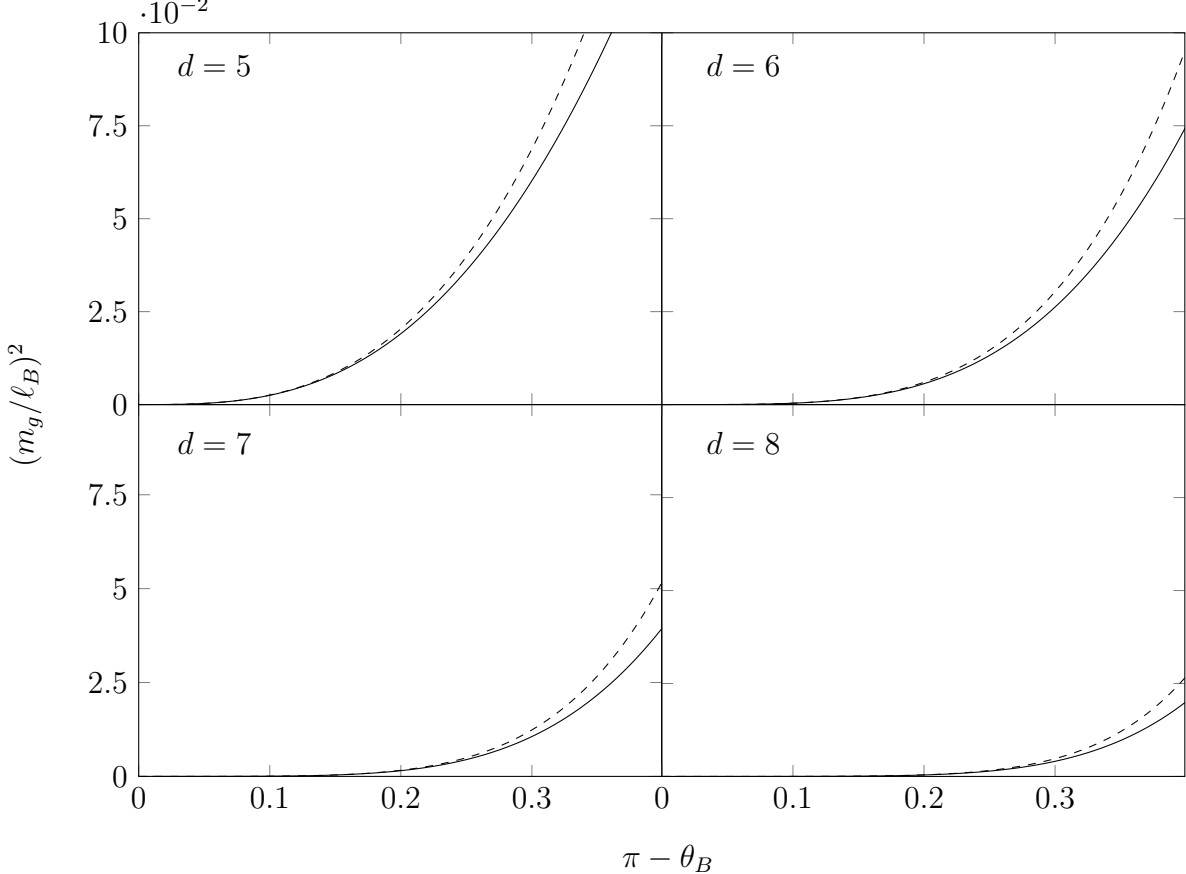

Figure 2: The graviton mass in various dimensions as a function of the location of the brane $\theta_B$. The solid line is the numerically determined value, while the dashed line is computed using eq. (61).

hole in equilibrium with thermal radiation, by placing a topological black hole into the bulk. The intersection of its horizon with the brane is the location of a black hole horizon on the brane. Seen from the brane perspective, such a setup exhibits an information paradox [4] and so one might have wanted to argue that firewalls appear (at least) past the Page time. However, we do not have a similar paradox in the bulk, since the radiation is reflected back into the black hole. As we have seen in this paper, the metric in the bulk determines the stress energy tensor on the brane. Thus, since the former has no reason to be singular at the horizon, the latter also should not be singular. Either way, the information paradox is resolved when using the island formula to compute entropies and so the need for a firewall might not arise to begin with.

In the second part of the paper we used this dictionary to derive a formula for the mass of the graviton in the brane theory. The question whether graviton masses are important for the island formula has been discussed in [5, 53, 54]. One interesting observation to make is that apart from the near boundary limit, the formula for the graviton mass exhibits

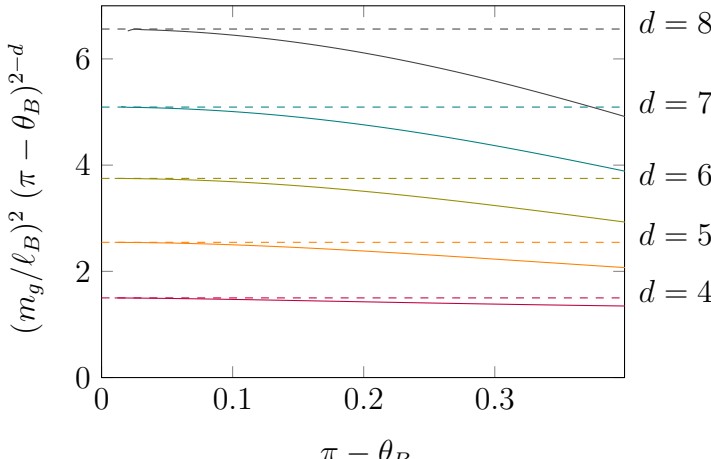

Figure 3: The squared (dimensionless) graviton mass divided by $(\pi - \theta_B)^{d-2}$. The solid line is computed numerically. One can see that the rescaled graviton mass approaches the analytically determined value given in table 1 as the brane angle becomes small (dashed line).

another limit in which the graviton mass vanishes, namely that of large dimension, $d \to \infty$. However, from the relation between Newtons constant in the bulk and the brane, we immediately see that in the same limit the gravitational coupling on the brane vanishes and thus islands become prohibitively expensive. One can already see this effect in [7], where the Page time

$$\tau_P \sim \frac{(d-1)^{\frac{d-1}{2}}}{(d-2)^{\frac{d}{2}}} \frac{1}{(\pi - \theta_B)^{d-2}} \tag{65}$$

diverges in the limit of infinite dimension. Proponents of the idea that islands are not possible without a graviton mass might interpret this as additional evidence. However, opponents might argue that the fact that it is possible that in the special class of Karch-Randall models the graviton mass is simply correlated with the gravitational coupling and thus sending the graviton mass to zero makes the formation of islands impossible. This however does not preclude the existence of models in which the graviton is massless and islands exist. In fact, arguments in effective field theory [34] seem to show that—if one allows for topology change—islands can exist in theories with massless gravitons.

Our presentation has focused on the Karch-Randall model and one might wonder to which extend the discussion carries over to the situation of cosmological spacetimes modeled using end-of-the-world branes [8, 14]. Cosmological models can be built by using a brane to cut off a region behind the horizon of an eternal black hole. If the tension of those branes can be chosen close to a critical value [9], the standard bulk and boundary descriptions of the system are again complemented by a third perspective, in which the system is approximately described by a CFT on the asymptotic boundary and an effective

CFT coupled to gravity in a big-bang/big-crunch cosmology. To which extent a bulk-brane dictionary is useful in this situation depends on the model. For realistic models of cosmology one would of course like to have standard model fields on the cosmological spacetime. This is usually done by adding an action localized to the brane and it is unclear how to extract detailed information about these fields from the bulk. Nonetheless, we are hopeful that a further investigation of the bulk/brane dictionary yields a better understanding of how the brane theory is encoded into the CFT state.

There are several open questions which warrant further work. The mass formula for the graviton is only one example of mass formulas for all bulk fields. The mass-terms for scalars, fermions and vectors should be calculable using essentially the same methods. It would further be interesting to extend the analysis beyond leading order in the near brane expansion or to include interactions. Another direct extension of this work would consider the effect of couplings of bulk fields to the brane. Lastly, the fate of the unstable scalars discussed at the end of section 2.4 needs further examination.

Moreover, we hope that the present work will be useful for investigating the dictionary between the BCFT and the effective CFT in the brane picture and ultimately to elucidate the microscopic origin on the island rule in higher dimensions.

## Acknowledgements

This work has benefited from discussions and collaborations an related projects with Vincent Chen, Ji-Hoon Lee, Ignacio Reyes, Joshua Sandor, and Ashish Shukla. I am particularly indebted to Rob Myers for a thorough reading of the manuscript and many valuable comments. I acknowledge support by the Simons Foundation through the "It from Qubit" collaboration. Research at Perimeter Institute is supported in part by the Government of Canada through the Department of Innovation, Science and Economic Development Canada and by the Province of Ontario through the Ministry of Colleges and Universities.

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
