# Peer review of "The Dictionary for Double Holography and Graviton Masses in d Dimensions"

_SciPost Physics_

## Round 2 · Referee Report · Andreas Karch (Referee 1) · 2021-9-2

Strengths

Gives new proposal for how to calculate expectation values in brane world holography which is backed up by quantitative evidence.

Weaknesses

Some conceptual issues could be more clearly addressed, see report.

Report

The manuscript discusses holography in sub-critical Randall-Sundrum branes. As reviewed, such brane worlds admit 3 different equivalent descriptions, referred to as the "bulk", "brane" and "boundary" perspective in this work. While a detailed dictionary between bulk and boundary had been understood before, this work tries to pin down the dictionary between bulk and brane. The brane perspective has found many recent applications in the context of quantitative descriptions of black hole evaporation, and so pinning down this dictionary is indeed important.

The author gives a proposal for how to calculate expectation values, generalizing the well know extrapolate dictionary of standard holography to this brane world. He then uses this proposal to calculate corrections to the mass of the graviton due to matter effects, and reproduces well known results that had been obtained from the bulk perspective. It is good to see quantitative evidence that the proposal makes sense.

One concern about this work I have is conceptual. There are two interesting questions I would like to see answered that are almost certainly closely related. 1) With gravity being dynamical on the brane, expectation values of local operators are not physical observables. Usually some dressing to asymptotic infinity is required. In any case, as it stands it is not obvious what observable is actually being calculated. 2) The dictionary the author presents seems to rely on working in the near-critical tension limit, where the curvature radius on the brane is much larger than the curvature radius in the bulk and the graviton is light. Why? Is there no dictionary for the general case?

I suspect 1) and 2) are related, only in the near-critical limit can one easily define more or less local operators in the standard way, appealing to "semi-classics". I would very much like to see 1) and 2), and whether and if so how the author thinks they are connected, discussed in the note. I, and I assume other readers as well, would at least like to see the author's take on this question.

With at least a brief discussion of these points included, I believe the work can be published in Scipost.

Requested changes

1- Include a discussion to address the conceptual question outlined in the report 2- Fix eq (8). I believe as written this can not be correct. Tcrit and T0 are dimensionless quantities, their difference can not be much less than 1. I believe the correct version should be: 0 << (Tcrit - T0)/Tcrit << 1 (or any equivalent version of this). This is just a minor issue and does not affect the results, but it should be corrected.

---

## Round 2 · Referee Report · Anonymous (Referee 2) · 2021-11-3

Strengths

  1. Important physical result derived in a transparent fashion

Weaknesses

  1. Some of the wordy bits of the manuscript are a bit cumbersome to read, like the beginning of Section 3.1.

Report

I strongly recommend publication. This paper derives the graviton mass in Karch-Randall models, in the small mass limit. This is an important result. The paper is also clearly and pedagogically written, for the most part.

Requested changes

  1. It would be nice to say a couple of words about what X_{ij} is right after equation (1) in the introduction. It is mentioned later of course, but since it appears in the introduction, it warrants a couple of words.

  2. On top of page 4, it is not clear what specifically the author means by "First, we of course have the finite corrections as indicated." Is this referring to the O(eps^#) piece in equation (1)?

  3. It would be nice to draw the theta coordinate on the pictures in figure 1.

  4. Typo after equation (5) "Einstein-Hilbert actions". Actions -> action

  5. The last sentence of section 2.1 refers to "leading order", but it is not clear leading order in what quantity. It would be useful to say that.

  6. Typo on page 15. "boundary conditions at the asymptotic"

  7. Indices missing on \delta G in the inline equation in the text below equation (41).

  8. It would be nice to say in the caption of Table 1 that the result is to leading nonzero order in pi - theta_B.

  9. Presumably the author means theta_B "greater than" pi - 0.1 at the end of section 3, instead of less than.

---

## Editorial Decision

awaiting_resubmission